# Single-cell transcriptomics defines heterogeneity of epicardial cells and fibroblasts within the infarcted murine heart

Julia Hesse[1†], Christoph Owenier[1†], Tobias Lautwein[2], Ria Zalfen[1], Jonas F Weber[3], Zhaoping Ding[1], Christina Alter[1], Alexander Lang[4], Maria Grandoch[5], Norbert Gerdes[4], Jens W Fischer[5], Gunnar W Klau[3], Christoph Dieterich[6], Karl Köhrer[2], Jürgen Schrader[1]*

[1]Department of Molecular Cardiology, Medical Faculty and University Hospital Düsseldorf, Heinrich-Heine-University Düsseldorf, Düsseldorf, Germany; [2]Biologisch-Medizinisches-Forschungszentrum (BMFZ), Genomics & Transcriptomics Laboratory, Heinrich-Heine-University Düsseldorf, Düsseldorf, Germany; [3]Algorithmic Bioinformatics, Heinrich-Heine-University Düsseldorf, Düsseldorf, Germany; [4]Division of Cardiology, Pulmonology and Vascular Medicine, Medical Faculty and University Hospital Düsseldorf, Heinrich-Heine-University Düsseldorf, Düsseldorf, Germany; [5]Institute of Pharmacology and Clinical Pharmacology, Medical Faculty and University Hospital Düsseldorf, Heinrich-Heine-University Düsseldorf, Düsseldorf, Germany; [6]Section of Bioinformatics and Systems Cardiology, Klaus Tschira Institute for Integrative Computational Cardiology and Department of Internal Medicine III, University Hospital Heidelberg, Heidelberg, Germany

*For correspondence: schrader@uni-duesseldorf.de

[†]These authors contributed equally to this work

Competing interests: The authors declare that no competing interests exist.

**Abstract** In the adult heart, the epicardium becomes activated after injury, contributing to cardiac healing by secretion of paracrine factors. Here, we analyzed by single-cell RNA sequencing combined with RNA in situ hybridization and lineage tracing of Wilms tumor protein 1-positive (WT1[+]) cells, the cellular composition, location, and hierarchy of epicardial stromal cells (EpiSC) in comparison to activated myocardial fibroblasts/stromal cells in infarcted mouse hearts. We identified 11 transcriptionally distinct EpiSC populations, which can be classified into three groups, each containing a cluster of proliferating cells. Two groups expressed cardiac specification markers and sarcomeric proteins suggestive of cardiomyogenic potential. Transcripts of hypoxia-inducible factor (HIF)-1α and HIF-responsive genes were enriched in EpiSC consistent with an epicardial hypoxic niche. Expression of paracrine factors was not limited to WT1[+] cells but was a general feature of activated cardiac stromal cells. Our findings provide the cellular framework by which myocardial ischemia may trigger in EpiSC the formation of cardioprotective/regenerative responses.

## Introduction

Myocardial infarction (MI), still the most frequent cause of death in western societies, is associated with massive activation of cardiac fibroblasts, ultimately resulting in excessive accumulation of extra-cellular matrix (ECM) components that finally impair cardiac function (*Tallquist and Molkentin, 2017*). During development, the majority of cardiac fibroblasts are derived from the epicardium, which forms the thin outermost epithelial layer of all vertebrate hearts and exhibits extensive

developmental plasticity (*Cao and Poss, 2018*). A subset of epicardial cells undergoes epithelial to mesenchymal transition (EMT) and those epicardial progenitor cells can give rise to various cardiac cell types. In addition to cardiac fibroblasts, these include vascular smooth muscle cells and pericytes, which contribute to the coronary vasculature (*Cao and Poss, 2018*). Embryonic epicardial heterogeneity was recently studied at the single-cell level in the developing zebrafish heart and uncovered three epicardial populations, functionally related to cell adhesion, migration, and chemotaxis (*Weinberger et al., 2020*).

In the adult heart, the epicardium is a rather quiescent monolayer that becomes activated after MI by upregulating embryonic epicardial genes (*Quijada et al., 2020*; *Masters and Riley, 2014*). In the injured heart, epicardial cells form via EMT a multi-cell layer of epicardial stromal cells (EpiSC) at the heart surface that can reach a thickness of about 50–70 µm in mice (*van Wijk et al., 2012*). It is generally assumed that the activated epicardium recapitulates the embryonic program in generating mesenchymal progenitor cells, although there may be major molecular differences with respect to their embryonic counterpart (*Bollini et al., 2014*). On the functional side, adult EpiSC secrete paracrine factors that stimulate cardiomyocyte growth and angiogenesis (*Zhou et al., 2011*) and play a key role in post-MI adaptive immune regulation (*Ramjee et al., 2017*). When stimulated with thymosin β4, Wilms tumor protein 1-positive (WT1+) adult EpiSC can form cardiomyocytes; however, the rate of conversion is only small (*Smart et al., 2011*). Thus, the epicardium is a signaling center regulating cardiac wound healing and may have cardiogenic potential in the adult injured heart.

Despite its importance in cardiac repair, little is known on cell heterogeneity and molecular identifiers within the epicardial layer of the adult heart. Yet, this knowledge is essential to map the epicardial progeny and attribute meaningful functions. While the single-cell landscape of activated cardiac fibroblasts (activated cardiac stromal cells, aCSC) in the post-MI heart has been explored in detail (*Farbehi et al., 2019*; *Forte et al., 2020*), these studies did not assess epicardial heterogeneity because of a lack of specific identifiers. In a previous study, we have reported a novel perfusion-based technique (*Owenier et al., 2020*), which permitted the simultaneous isolation of EpiSC and aCSC with high yields and only minimal cell activation. The isolation of viable, purified preparations of aCSC and EpiSC from the infarcted heart permitted the first direct comparison of the two cardiac stromal cell fractions. We have combined single-cell RNA sequencing (scRNAseq) with lineage tracing of WT1-expressing cells and localization of cell populations by RNA in situ hybridization, characterized cellular hierarchy of adult EpiSC, defined similarities and differences to cardiac fibroblasts, and explored in detail the individual EpiSC populations in the activated epicardium.

## Results

### scRNAseq of post-MI stromal cells

EpiSC and aCSC were isolated from the same mouse hearts ($n$ = 3) 5 days after MI (50 min ischemia followed by reperfusion) by combined collagenase perfusion and bathing (*Figure 1A*). This recently developed technique selectively removes the majority of EpiSC from the aCSC-containing myocardium by applying gentle shear forces to the cardiac surface and has been validated for cell yield, viability, and purity (*Owenier et al., 2020*). Day 5 was chosen because the epicardial cell layer had reached its maximal dimension (*Ding et al., 2016*) and fetal epicardial genes as a measure of epicardial activation were re-expressed (*Zhou et al., 2011*). CSC were isolated from uninjured hearts of sham-operated animals ($n$ = 3) without separation of epicardial cells (see 'Materials and methods'). Cell preparations were depleted of cardiomyocytes, endothelial cells, and immune cells prior to scRNAseq using the 10x Genomics Chromium platform. Transcriptional profiles of 13,796 EpiSC, 24,470 aCSC, and 24,781 CSC were captured after quality control filtering. Unbiased clustering using the Seurat R package with visualization in uniform manifold approximation and projection (UMAP) dimension reduction plots was performed to identify cells with distinct lineage identities and transcriptional profiles. Average and significantly enriched RNA expression of EpiSC, aCSC, and CSC are listed in *Supplementary files 1* and *2*, respectively.

### Characterization of EpiSC populations

As shown in *Figure 1B*, we identified 11 transcriptionally different cell populations within the EpiSC fraction. Cell doublets with hybrid transcriptomes identified by DoubletFinder (*McGinnis et al.,*

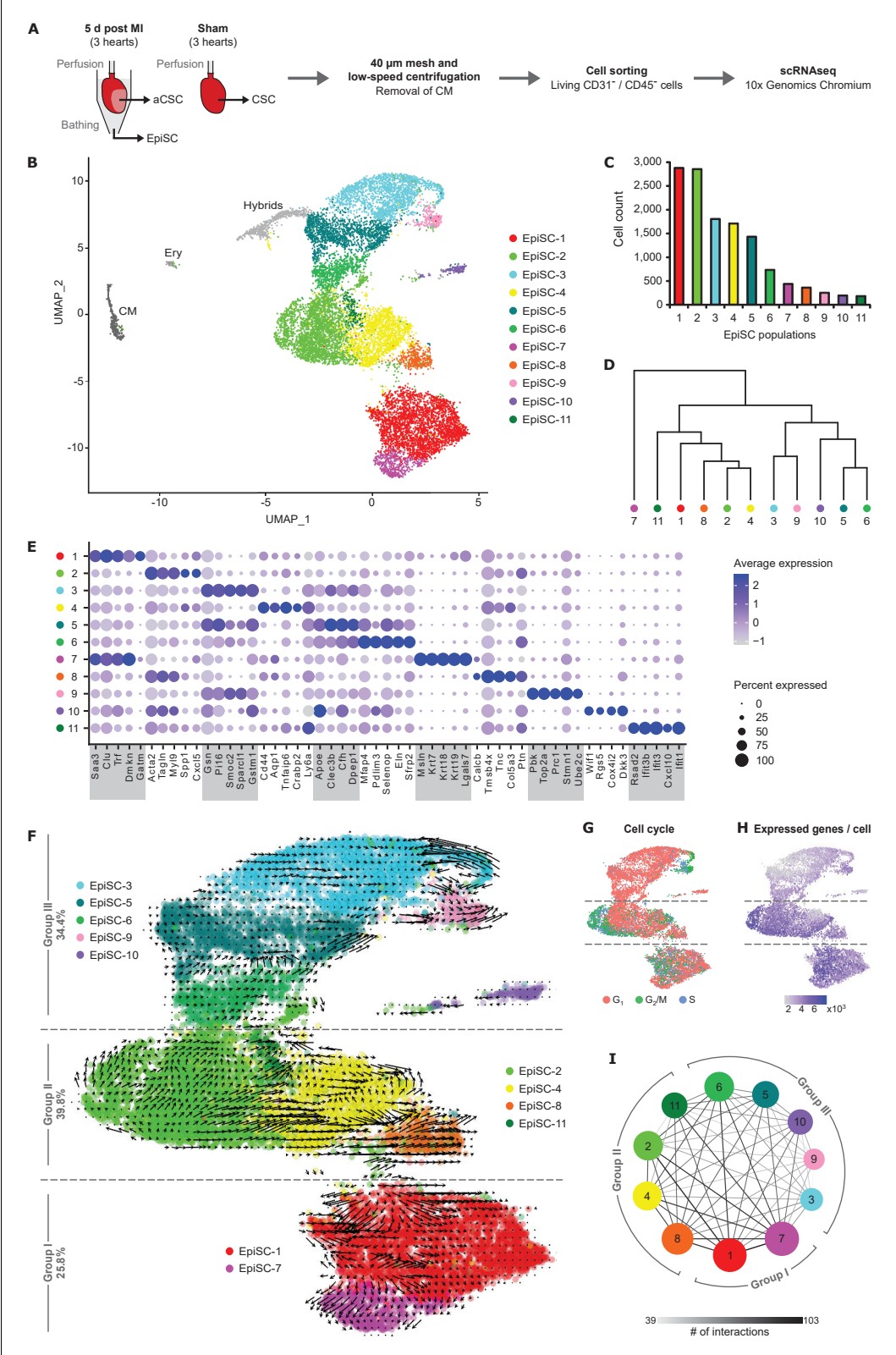

**Figure 1.** Cell populations in EpiSC from the infarcted heart. (**A**) Schematic workflow. Epicardial stromal cells (EpiSC) and activated cardiac stromal cells (aCSC) were simultaneously collected from the surface and the myocardium of the isolated perfused heart by applying mild shear forces to the cardiac surface (**Owenier et al., 2020**) at 5 days post myocardial infarction (MI) (*n* = 3). CSC were purified from three non-infarcted control hearts 5 days after sham surgery. Mesh purification, low-speed centrifugation, and cell sorting by flow cytometry were performed to remove cardiomyocytes, *Figure 1 continued on next page*

*Figure 1 continued*

CD31⁺ endothelial cells, CD45⁺ immune cells, and apoptotic or necrotic cells before analysis using the 10x Genomics Chromium platform. (**B**) UMAP plot of clustered single-cell RNA sequencing (scRNAseq) data of the pooled EpiSC fraction ($n = 13,796$ single cells). CM, cardiomyocytes; Ery, erythrocytes. (**C**) Cell count of EpiSC populations. (**D**) Dendrogram of EpiSC populations according to average RNA expression. (**E**) Dot plot of top five marker genes for each EpiSC population. (**F**) RNA velocity of EpiSC populations projected on the UMAP plot. Arrows show the local average velocity evaluated on a regular grid, indicating estimated future states. (**G**) Feature plot displaying the predicted classification in $G_1$, $G_2$/M, and S phase for each cell according to cell cycle gene expression scores. (**H**) Feature plot displaying the number of unique genes detected in each cell. (**I**) Networks visualizing the number of potential specific interactions between EpiSC populations as determined by CellPhoneDB.

The online version of this article includes the following source data and figure supplement(s) for figure 1:

**Source data 1.** Source data for EpiSC population cell counts summarized in *Figure 1C*.
**Figure supplement 1.** Excluded hybrid and non-stromal cell populations in the EpiSC fraction.
**Figure supplement 2.** Gene ontology terms typifying EpiSC populations.
**Figure supplement 3.** Top 10 marker genes of EpiSC populations.
**Figure supplement 4.** Receptor–ligand pairs in EpiSC populations.

*2019*; *Figure 1—figure supplement 1A*) and minor non-stromal cell populations such as cardiomyocytes and erythrocytes, identified according to cell type-specific marker expression (*Figure 1—figure supplement 1B*), were excluded from further analysis. EpiSC populations varied in size (*Figure 1C*), were hierarchically structured (*Figure 1D*), and the differentially expressed genes showed over-representations of distinct gene ontology (GO) biological process terms within each population (*Figure 1—figure supplement 2*). Differentially expressed population marker genes are summarized as top five dot plot in *Figure 1E* and top 10 heat map in *Figure 1—figure supplement 3*. Remarkably, transcripts of epithelial cell-associated genes and established epicardial genes (*Bochmann et al., 2010*) were enriched in both EpiSC-1 (*Dmkn*, *Saa3*) and EpiSC-7 (*Msln*, *Krt7*, *Krt18*, *Krt19*, *Lgals7*), while mesenchymal marker genes were primarily expressed in EpiSC-3 (*Gsn*, *Pi16*) and EpiSC-4 (*Cd44*, *Ly6a*). Genes coding for ECM proteins were highly expressed in EpiSC-2 (*Spp1*), EpiSC-3 (*Smoc2*, *Sparcl1*), EpiSC-5 (*Clec3b*), EpiSC-6 (*Mfap4*, *Eln*), and EpiSC-8 (*Col5a3*, *Tnc*). Genes encoding contractile proteins were preferentially expressed in EpiSC-2 (*Acta2*, *Myl9*, *Tagln*). Wnt pathway-associated gene transcripts were enriched in EpiSC-6 (*Sfrp2*) and EpiSC-10 (*Wif1*, *Dkk3*). Genes related to the cellular response to interferon characterized EpiSC-11 (*Ifit1*, *Ifit3*, *Ifit3b*, *Cxcl10*). Finally, expression of genes associated with high cell cycle activity and mitosis was a feature of EpiSC-9 (*Pbk*, *Top2a*, *Prc1*, *Stmn1*, *Ube2c*).

## Hierarchy of EpiSC populations

To better define the cellular relationship between the identified EpiSC populations, we performed RNA velocity analysis (*La Manno et al., 2018*), which can predict and visualize future cell states based on the ratio of unspliced and spliced mRNA read counts. As shown in *Figure 1F*, this trajectory inference separated the EpiSC populations into three groups: EpiSC-7 and -1 (group I), EpiSC-2, -4, -8, -11 (group II), and EpiSC-6, -5, -3, -9, -10 (group III).

Cell cycle scoring showed that each of the three EpiSC groups identified above was equipped with a cell cluster expressing a number of $G_2$/M and S phase genes (*Figure 1G*), indicating proliferating cells: within group I, a subpopulation of EpiSC-1, within group II, a subpopulation of EpiSC-2, and within group III, EpiSC-9. This location is very similar to the origin of velocity arrows (*Figure 1F*).

The number of expressed genes per cell was lower in group III than in groups I and II (*Figure 1H*). Since the expressed genes per cell correlate with developmental potential (*Gulati et al., 2020*), this may indicate that a large fraction of group III cells are terminally differentiated.

To study cell–cell communication mediated by ligand–receptor interactions between the EpiSC populations, we used CellPhoneDB (*Efremova et al., 2020*). As shown in *Figure 1I*, the highest numbers of interactions were predicted between groups I and II. Ligand–receptor pairs potentially involved in the interaction between the groups include IGF1, transforming growth factor beta-2 (TGF-β2), and Wnt4 signaling (*Supplementary file 3*, *Figure 1—figure supplement 4*).

## Epicardial marker gene expression

The cellular distribution of well-established epicardial progenitor marker genes such as *Wt1*, *Tbx18*, *Sema3d, Aldh1a2, Gata5*, and *Tcf21* (*Cao and Poss, 2018*) is shown in *Figure 2A*. *Wt1*, commonly used for lineage tracing studies (*Quijada et al., 2020*), was found to be highly enriched in group I EpiSC populations 1 and 7, as well as the adjacent population EpiSC-8. *Sema3d, Aldh1a2*, and *Gata5* were also predominantly expressed in group I, with the highest expression levels in EpiSC-7. *Tbx18*, on the other hand, was broadly distributed in group I and II populations EpiSC-1, -2, -4, -7, -8, -9, and -11 (*Figure 2A*). In contrast, *Tcf21* was strongly expressed in group III EpiSC-3, -5, -6, and -9 and showed no overlap with *Wt1*-expressing populations (*Figure 2A*).

Since cells in EpiSC-1 showed substantial inhomogeneity of markers (*Figure 2A*), we carried out a separate clustering analysis for EpiSC-1, including the two adjacent *Wt1*-expressing populations EpiSC-7 and -8. We identified five subclusters in EpiSC-1 (EpiSC-1.1, -1.2, -1.3, -1.4, -1.5), with no subclustering in EpiSC-7 and -8 (*Figure 2—figure supplement 1A–C*). EpiSC1.4 is characterized by expression of cell cycle-associated genes (*Rrm2, Pclaf, Hist1h2ap, Hmgb2, Ube2c, Top2a*) (*Figure 2—figure supplement 1D*; *Supplementary files 4* and *5*) and correlates to the proliferative EpiSC-1 subpopulation identified by cell cycle scoring (*Figure 1G*). EpiSC1.5 showed enriched expression of genes encoding core ribosomal proteins such as *Rps21* and *Rpl37a* (*Figure 2—figure supplement 1D*; *Supplementary file 5*), suggesting high protein synthesis activity.

We also searched for expression of *Tgm2, Sema3f*, and *Cxcl12*, which were recently found to mark three functional different epicardial populations in the developing zebrafish heart (*Weinberger et al., 2020*). We found *Tgm2* and *Sema3f* preferentially expressed in group I populations EpiSC-1 and -7 (*Figure 2—figure supplement 2A–B*). However, *Mylk*, which is an additional marker of the *Sema3f*-expressing zebrafish epicardial population (*Weinberger et al., 2020*), was primarily expressed in EpiSC-8. *Cxcl12*, present only in a small cell population in the zebrafish (*Weinberger et al., 2020*), was rather broadly expressed with enrichment in EpiSC-2 and subcluster EpiSC-1.1 (*Figure 2—figure supplement 2C–D*). These findings not only demonstrate some degree of evolutionary preservation in epicardial populations from zebrafish to mice, but also reveal considerable differences in marker gene expression patterns and population sizes.

## Spatial mRNA expression of EpiSC population identifiers

To determine the specific location of EpiSC populations in the infarcted heart, RNA in situ hybridization of gene transcripts for selected EpiSC populations was carried out (*Figure 2B*, upper panel), using *Wt1* and other population-specific identifiers (*Figure 2B*, lower panel). *Msln* expression (EpiSC-7) was detected on the outmost layers of the epicardium, consistent with the epithelial signature of EpiSC-7. Expression of *Wt1* (EpiSC-7, -1, -8) and *Cd44* (EpiSC-4) similarly labeled cells that were localized in the outer part. In contrast, Ifit3 (EpiSC-11) expression was also found in deeper layers of the activated epicardium and was additionally detected in the myocardium, labeling aCSC. Expression of group III population markers *Sfrp2* (EpiSC-6), *Dkk3* (EpiSC-10), and *Pcsk6* (EpiSC-3) was rather homogenously dispersed throughout the epicardium and the aCSC in the infarct border zone. *Top2a*, chosen as an identifier of the proliferative (sub)populations of all three groups (EpiSC-1.4, EpiSC-2 subpopulation, EpiSC-9), showed a similar expression pattern. This scattered distribution of proliferating cells in the post-MI heart and especially within the epicardial multi-cell layer has already been confirmed in BrdU incorporation experiments (*van Wijk et al., 2012*).

To explore whether the dispersed *Wt1*-negative group II and group III cells might derive from the *Wt1*-expressing populations located in the outmost layer of the activated epicardium, we combined scRNAseq of EpiSC with lineage tracing using tamoxifen-inducible *Wt1*-targeted reporter (*Wt1$^{CreER-T2}$Rosa$^{tdTomato}$*) mice (*Figure 2C*). Transcriptional profiles of 13,373 cells from two mouse hearts 5 days after MI were assigned to previously defined EpiSC populations (*Figure 2D*, *Figure 2—figure supplement 3*). Expression of tdTomato fully overlapped with that of *Wt1* (*Figure 2E*), suggesting that during the first days of MI-triggered epicardial activation and expansion, WT1$^{+}$ cells did not contribute to cells of *Wt1*-negative EpiSC populations.

**Figure 2.** Molecular characterization and location of EpiSC populations. (**A**) Expression of epicardial progenitor cell markers proposed in the literature as visualized in feature and violin plots. (**B**) Upper panel: RNA in situ hybridization of epicardial stromal cell (EpiSC) population identifiers (red) in heart cryosections 5 days post myocardial infarction (MI). Representative images (*n* = 4 hearts) of the infarct border zone are shown. The dotted line marks the interface between myocardial/epicardial tissue according to cell morphology. Nuclei were stained with hematoxylin (blue). Scale bars, 50 µm. Lower

*Figure 2 continued on next page*

*Figure 2 continued*

panel: feature plots visualizing selected EpiSC population molecular identifiers and scheme of the post-MI heart with the infarct zone in grey. RV, right ventricle; LV, left ventricle. (C–E) Lineage tracing of *Wt1*-expressing cell populations post MI using *Wt1^CreERT2^Rosa^tdTomato^* mice. The experimental design is outlined in (C). UMAP plot of clustered scRNAseq data of the EpiSC fraction (*n* = 13,373 single cells) pooled from two hearts 5 days post MI is shown in (D). Expression of *Wt1* and tdTomato is visualized in (E).

The online version of this article includes the following figure supplement(s) for figure 2:

**Figure supplement 1.** Subclusters of *Wt1*-expressing EpiSC populations.
**Figure supplement 2.** Expression of epicardial markers previously identified in the developing zebrafish heart in mouse post-MI EpiSC.
**Figure supplement 3.** *Wt1*-expressing subclusters in *Wt1^CreERT2^Rosa^tdTomato^* EpiSC.

## Expression of cardiomyogenesis-associated genes, HIF-1-responsive genes, and paracrine factors

Lineage tracing experiments have shown that cardiomyogenesis can be initiated in vivo in WT1⁺ epicardial cells when stimulated with thymosin β4 (10). We found thymosin β4 (*Tmsb4x*) highly expressed in *Wt1*-expressing EpiSC-8 (*Figure 3A and B*), together with BRG1 (*Smarca4*), a transcription activator for WT1 (*Vieira et al., 2017*). Interestingly, many key cardiogenic factors were also expressed within the *Wt1*-expressing EpiSC populations (*Figure 3A*, cardiogenic factors I). These include MESP1, which marks early cardiovascular progenitor specification (*Chiapparo et al., 2016*), as well as GATA4 (*Figure 3B*), which in combination with MEF2C and TBX5 has been shown to facilitate cardiomyogenesis (*Ieda et al., 2010*). Surprisingly, several *Hox* family members were expressed in EpiSC-8, for example, *Hoxa5* (*Figure 3A and B*). HOX transcription factors are downstream effectors of retinoic acid signaling (*Nolte et al., 2019*), which is required for differentiation of cardiac progenitors during heart development (*Roux and Zaffran, 2016*). Intriguingly, a second set of cardiogenic factors was found in group III populations EpiSC-3, -5, -6, -9, and -10 (*Figure 3A*, cardiogenic factors II). This includes MEF2C (*Figure 3B*), Nkx-2.5, as well as BMP2 and BMP4, the latter of which are crucial in the regulation of Nkx-2.5 expression and specification of the cardiac lineage (*Brown et al., 2004*). In addition, we found low expression of genes encoding muscle structural proteins such as *Myl2*, -4, and -6, *Tnnt2*, *Ttn,* and *Nebl*, which appear to be present especially in EpiSC-7 and EpiSC-9 (*Figure 2C*, contractile proteins). The highest expression levels of Notch target genes were found in *Wt1*-negative EpiSC populations, especially EpiSC-10 (*Figure 3A*). This is remarkable, since Notch-activated epicardial-derived WT1⁺ cells were described as a multipotent cell population with the ability to express cardiac genes (*Russell et al., 2011*). The observation that EpiSC-7 and -9 express cardiac specification markers and sarcomere proteins is suggestive that these populations have cardiomyogenic potential.

A hypoxia-responsive element in the *Wt1* promoter was reported to bind HIF-1α, which is required for WT1 induction (*Wagner et al., 2003*). We found expression of *Hif1a* and HIF-1-responsive genes, particularly those associated with glycolysis, to be enriched in group I and group II EpiSC populations (*Figure 3C*). These include the lactate-generating enzyme lactate dehydrogenase (*Ldha*) and the lactate-exporting monocarboxylate transporter 4 (*Slc16a3*) (*Figure 3C and D*). Consistently, the genes with significantly higher expression in group I and II populations vs group III populations (*Supplementary file 6*) showed enrichment of several GO terms related to the hypoxia/HIF-mediated metabolic switch toward oxygen-saving glycolysis, such as glycolytic process (FDR q-value: 3.05E-02), glucose metabolic process (FDR q-value: 3.22E-02), and pyruvate biosynthetic process (FDR q-value: 2.66E-02). Due to the transient nature of the HIF-1 response that strongly depends on local oxygen levels, the enriched expression of HIF-1-responsive genes in group I and group II in comparison to group III suggests that the populations reside in microenvironments with different oxygen levels within the post-MI epicardium.

Epicardial cells have been reported to secrete numerous paracrine factors that can modulate myocardial injury in the mouse heart (*Zhou et al., 2011*). In line with this observation, we found enriched expression of various chemokines in group II EpiSC populations (*Figure 3E*). Several chemokines known to be involved in monocyte recruitment were primarily expressed in EpiSC-11, including MCP1 (*Ccl2*) and MCP3 (*Ccl7*) (*Figure 3E and F*). EpiSC-11 also showed strongly enriched expression of IP-10 (*Cxcl10*), which is involved in triggering anti-fibrotic effects after MI (*Bujak et al., 2009*). Expression of chemokines involved in neutrophil granulocyte recruitment (GRO-α, -β, -γ,

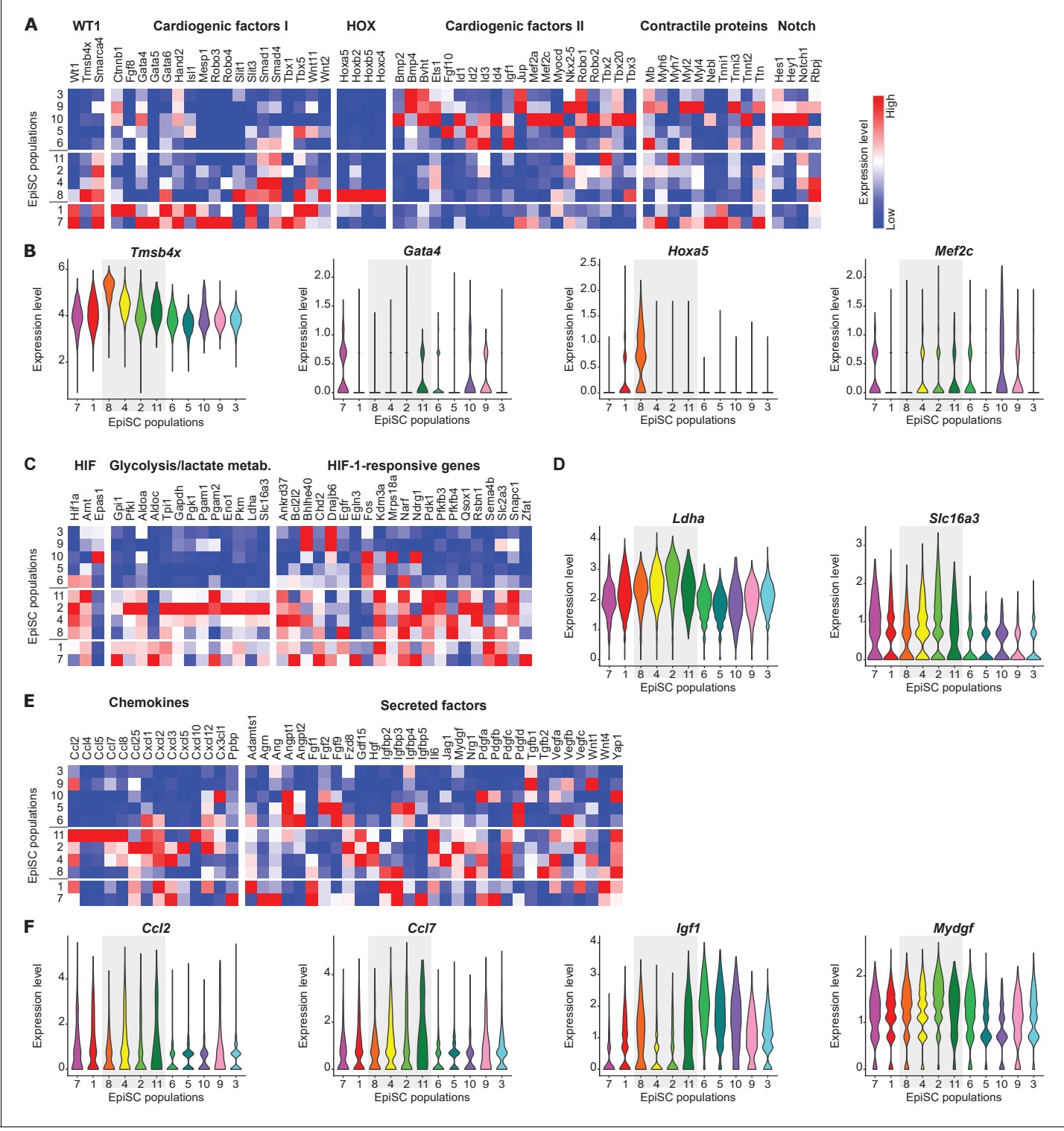

**Figure 3.** Functional characterization of EpiSC populations. (A, B) Heat maps (A) and violin plots (B) of selected genes showing the expression of cardiogenic factors and Notch target genes. (C, D) Heat maps (C) and violin plots (D) of genes associated with HIF-1 signaling. (E, F) Heat maps (E) and violin plots (F) of genes encoding chemokines and further secreted factors.

*Cxcl1, -2, -3*; ENA-78, *Cxcl5*; NAP-2/CXCL-7, *Ppbp*) was preferentially found not only in EpiSC-2 and -4 but also in EpiSC-1 and -7. Expression of SDF-1 (*Cxcl12*), known to reduce scar size when administered to the damaged heart (*Sundararaman et al., 2011*), was enriched in EpiSC-2.

Expression of proangiogenic factors previously identified in the supernatant of WT1⁺ epicardial cells (*Zhou et al., 2011*), such as Angiopoietin-1 (*Angpt1*), FGF2, IL-6, VEGFA, and VEGFC, was not limited to the *Wt1*-expressing EpiSC populations but pertained to other identified EpiSC populations (*Figure 3E*). The most highly expressed paracrine factors were TGF-β1, which attenuates myocardial ischemia-reperfusion injury (*Chen et al., 2003*), IGF-1, which prevents long-term left ventricular remodeling after cardiac injury (*Dai and Kloner, 2011*; *Figure 3F*), and MYDGF, which mediates ischemic tissue repair (*Korf-Klingebiel et al., 2015*; *Figure 3F*).

## Post-MI aCSC in comparison to EpiSC

Clustering analysis of aCSC revealed 11 transcriptionally different populations (*Figure 4—figure supplement 1A–C*). Again, cell doublets with hybrid transcriptomes identified by DoubletFinder (*McGinnis et al., 2019*; *Figure 4—figure supplement 2A*) and minor non-stromal cell populations (*Figure 4—figure supplement 2B*) were excluded from further analysis. A dot plot of the top five and a heat map of the top 10 most differentially expressed genes in aCSC populations are displayed in *Figure 4—figure supplement 1D* and *Figure 4—figure supplement 3*, respectively. To underline the segregation of aCSC populations with partially overlapping marker genes (aCSC-4, -5, and aCSC-7, -10, respectively), the expression of selected population marker genes is also visualized in feature plots (*Figure 4—figure supplement 1E*). Genes encoding ECM proteins were associated with aCSC-1 (*Eln*, *Wisp2*), aCSC-2 (*Thbs4*, *Cilp*), and aCSC-4 (*Smoc2*, *Sparcl1*). Cell populations with similar gene signatures were recently termed activated fibroblasts (*Farbehi et al., 2019*) or late-response fibroblasts and matrifibrocytes (*Forte et al., 2020*). In these studies, a population similar to aCSC-5 was referred to as either Sca1-low or homeostatic epicardial-derived fibroblasts (*Farbehi et al., 2019*; *Forte et al., 2020*). aCSC-6 preferentially expressed genes characteristic of epithelial cells (*Lgals7*, *Dmkn*, *Msln*). Because this epithelial signature was comparable to that observed for EpiSC-7, it is very likely that cells of aCSC-6 are of epicardial origin and are included in the aCSC fraction due to an incomplete removal of the EpiSC layer from the myocardium as to be expected from the used cell isolation technique (*Owenier et al., 2020*). Consistent with this, in situ hybridization identified expression of the aCSC-6 and EpiSC-7 marker *Msln* exclusively within the epicardium (*Figure 2B*). Similar to so-called cycling fibroblasts/proliferating myofibroblasts (*Farbehi et al., 2019*; *Forte et al., 2020*), we found genes involved in cell cycle and mitosis preferentially in aCSC-7 (*Cenpa*, *Stmn1*, *Ccnb2*) and aCSC-10 (*Top2a*, *Ube2c*, *Hist1h2ap*, *Pclaf*). Again similar to data in the literature (*Farbehi et al., 2019*; *Forte et al., 2020*), we found genes related to the cellular response to interferon highly expressed in aCSC-8 (*Ifit3*, *Isg15*, *Ifit1*, *Iigp1*, *Ifit3b*), which were termed interferon-stimulated/interferon-responsive fibroblasts. aCSC-9 highly expressed *Ly6a*, encoding Sca1, and resembled the reported Sca1-high/progenitor-like fibroblasts (*Farbehi et al., 2019*; *Forte et al., 2020*). Genes encoding contractile proteins were preferentially expressed in aCSC-3 (*Acta2*, *Tpm2*), which were recently referred to as myofibroblasts (*Forte et al., 2020*). Expression of Wnt pathway-associated genes was enriched in aCSC-1 (*Sfrp1*, *Sfrp2*) and aCSC-11 (*Wif1*, *Dkk3*), the latter of which were referred to as Wnt-expressing/endocardial-derived fibroblasts (*Farbehi et al., 2019*; *Forte et al., 2020*). Taken together, all aCSC populations identified by us are consistent with previously identified cardiac fibroblast populations.

To directly compare the transcriptional profile of EpiSC to myocardial aCSC, we performed canonical correlation analysis (CCA) space alignment of these two post-MI scRNAseq data sets, together with CSC from uninjured hearts. Generated CCA clusters (*Figure 4A*, left panel) are also displayed according to original cell IDs (*Figure 4A*, right panels). Average RNA expression levels and differentially expressed marker genes of CCA clusters are listed in *Supplementary files 7* and *8*, respectively. The contribution of EpiSC, aCSC, and CSC fractions to CCA clusters is summarized in *Figure 4B*. As can be seen, CCA clusters A, D, and F were mainly composed of EpiSC, indicating that the transcriptional profile of EpiSC-1, -4, -7, and -8 is prevalent in the epicardium (*Figure 4B*). On the other hand, CCA clusters B, C, H, I, J, and O were dominated by aCSC. Individual assignments showed that *Wt1*-negative EpiSC-2, -6, and -9 carried to variable degree expression signatures of genuine aCSC-1, -2, -3, -4, -5, and -10. Despite these similarities, there were multiple significant differences in average gene expression levels when comparing *Wt1*-negative EpiSC and

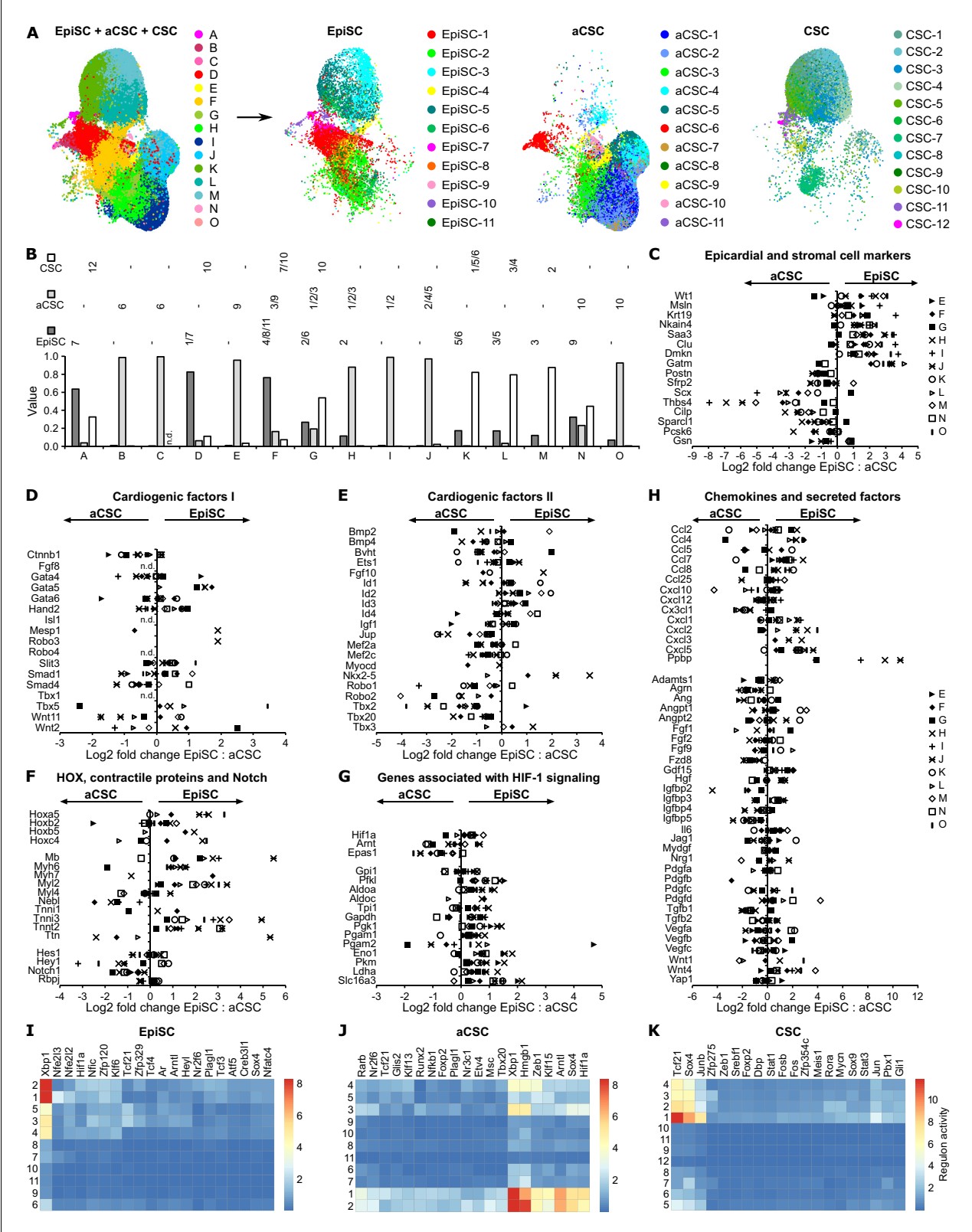

**Figure 4.** Comparison of EpiSC, aCSC, and CSC. (**A**) Canonical correlation analysis (CCA) space alignment of epicardial stromal cells (EpiSC), activated cardiac stromal cells (aCSC), and CSC single-cell RNA sequencing (scRNAseq) data in one UMAP plot (left) and split in one plot each (right). Cells are color-coded according to their assignment to CCA clusters (left) or previously identified populations (right). (**B**) Contribution of EpiSC, aCSC, and CSC fractions to CCA clusters. n.d., not detected. (**C–H**) Comparison of EpiSC and aCSC gene expression profiles in CCA clusters D-O regarding epicardial

*Figure 4 continued on next page*

*Figure 4 continued*

and stromal cell markers (**C**), cardiogenic factors (**D and E**), HOX transcription factors, contractile proteins and Notch target genes (**F**), genes associated with HIF-1 signaling (**G**) as well as chemokines and other secreted factors (**H**) as log2 fold change. n.d., not defined. (**I–K**) Gene regulatory network analysis in EpiSC (**I**), aCSC (**J**), and CSC (**K**) populations by SCENIC.

The online version of this article includes the following source data and figure supplement(s) for figure 4:

**Figure supplement 1.** Cell populations in aCSC from infarcted myocardium.
**Figure supplement 1—source data 1.** Source data for aCSC population cell counts summarized in *Figure 4—figure supplement 1B*.
**Figure supplement 2.** Excluded hybrid and non-stromal cell populations in the aCSC fraction.
**Figure supplement 3.** Top 10 marker genes of aCSC populations.
**Figure supplement 4.** Cell populations in CSC from uninjured hearts.
**Figure supplement 4—source data 1.** Source data for CSC population cell counts summarized in *Figure 4—figure supplement 4B*.
**Figure supplement 5.** Excluded hybrid and non-stromal cell populations in the CSC fraction.
**Figure supplement 6.** Top 10 marker genes of CSC populations.
**Figure supplement 7.** Comparison of EpiSC to CSC.
**Figure supplement 7—source data 1.** Source data for differentially expressed genes among EpiSC and CSC in CCA cluster A displayed in *Figure 4—figure supplement 7D*.

aCSC within individual CCA clusters (*Supplementary file 7*). *Wt1*-expressing CCA clusters A–D were excluded from this comparative analysis, since *Wt1* expression was only found at the heart's surface (*Figure 2B*), strongly suggesting that aCSC assigned to these clusters probably represent epicardial contamination in the aCSC fraction (see above). As shown in *Figure 4C*, expression of epithelial/epicardial genes was generally higher in EpiSC, while mesenchymal/fibroblast genes were more dominant in aCSC. A distinct fraction of cardiogenic factors set I (see *Figure 3A*) was preferentially expressed in EpiSC (*Figure 4D*) while the opposite was true for cardiogenic factors set II (*Figure 4E*). Notably, cardiac contractile proteins (*Figure 4F*) and HIF-1-responsive glycolytic enzymes (*Figure 4G*) were predominantly expressed within EpiSC. Among the paracrine factors (*Figure 4H*), we found multiple chemokines highly enriched in EpiSC.

To further compare cell states in our stromal cell fractions, we used SCENIC (*Aibar et al., 2017*) for gene regulatory network reconstruction. This tool scores the activity of transcription factors by correlating their expression with the expression of their direct-binding target genes (*Aibar et al., 2017*). As shown in *Figure 4I and J*, EpiSC and aCSC showed distinct patterns of network activity, again emphasizing their different cellular identities and functions. As to be expected, both post-MI fractions showed major differences in network activity compared to CSC from uninjured hearts (*Figure 4K*).

## Comparison to CSC from uninjured hearts

Clustering analysis of CSC revealed 12 transcriptionally different populations (*Figure 4—figure supplement 4A–C*) after cell doublets (*Figure 4—figure supplement 5A*) and minor non-stromal cell populations (*Figure 4—figure supplement 5B*) were excluded. Interestingly, the smallest CSC population (CSC-12) highly expressed epithelial/epicardial genes (*Msln*, *Upk3b*, *Nkain4*, *Krt19*) (*Figure 4—figure supplement 4D*, *Figure 4—figure supplement 6*) and most likely represents cells of the epicardial monolayer of the uninjured heart. Individual single-cell analysis of this monolayer is technically not feasible, because the number of cells liberated by shear forces from uninjured hearts is only rather small (estimated ~5600 cells/heart) and there is shear-independent release of cells (background) of unknown origin (*Owenier et al., 2020*). CSC-12 was the only *Wt1*-expressing population in the CSC fraction (*Figure 4—figure supplement 4E*).

As shown in *Figure 4A–B*, CCA space alignment of the CSC and EpiSC data sets revealed that the majority of CSC correlated with EpiSC group III populations EpiSC-3, -5, and -6 (CCA cluster K, L, M). Direct comparison of EpiSC and CSC within individual CCA clusters again showed that expression of epithelial/epicardial genes was enriched in EpiSC, while transcript levels of the conventional fibroblast marker *Gsn* were higher in CSC (*Figure 4—figure supplement 7A*). EpiSC showed a generally higher expression of HIF-1-responsive genes (*Figure 4—figure supplement 7B*) and proliferation-associated genes (*Figure 4—figure supplement 7C*), consistent with the notion of increased epicardial hypoxia and induction of epicardial cell proliferation in the infarcted heart. Indeed, direct comparison of the epithelial populations EpiSC-7 and CSC-12 in CCA cluster A revealed that HIF-1-

responsive/glycolysis-associated genes (*Eno1*, *Gale*, *Ldha*, *Slc16a3*) and proliferation-associated genes (*Calm*, *Cenpe*, *Ran*, *Ranbp1*, *Spc25*, *Ube2c*) are upregulated in post-MI EpiSC-7 (*Figure 4— figure supplement 7D*).

## Discussion

In this study, we provide a single-cell landscape of the post-MI epicardium with 11 transcriptionally different cell populations. This amazing degree of cellular heterogeneity is similar to that of myocardial cardiac fibroblasts (*Farbehi et al., 2019*; *Forte et al., 2020*). The widely used epicardial lineage marker gene *Wt1* (*Quijada et al., 2020*) was selectively expressed in three populations (EpiSC-1, -7, -8), which were localized on the outer surface of the activated epicardium (*Figure 2B*). Other commonly used lineage markers and recently identified epicardial population markers of the developing zebrafish heart (*Weinberger et al., 2020*) were rather heterogeneously distributed (*Figure 2A*, *Figure 2—figure supplement 2A–B*), suggesting that they mark different stages of differentiation, commitment, or activity in the adult heart.

The quality of scRNAseq critically depends on the cell isolation technique, which ideally should preserve the native state of cells as close as possible. To minimally perturb the native expression profile of cardiac stromal cells, we used a recently elaborated perfusion protocol for the simultaneous isolation of EpiSC and aCSC from the same infarcted heart, which is short (8 min) and results in a high yield of viable cells (*Owenier et al., 2020*). The application of mild shear forces on the cardiac surface by a simple motor-driven device permitted the rather selective removal of the EpiSC fraction (*Owenier et al., 2020*). As compared to a commonly used mincing protocol (30–40 min), we found the yield of aCSC to be higher with our technique, and this was associated with significant lower induction of immediate early response genes (*Owenier et al., 2020*). Because of these reasons, our data are difficult to compare with published studies at single-cell resolution of the post-MI heart, which relied on a mincing protocol. These studies have identified either no (*Skelly et al., 2018*; *Gladka et al., 2018*; *Kretzschmar et al., 2018*) or only one (*Farbehi et al., 2019*; *Forte et al., 2020*; *Cui et al., 2019*; *Asp et al., 2019*) epicardial cell population.

RNA velocity analysis (*Figure 1F*) and cell cycle scoring (*Figure 1G*) suggest that the EpiSC populations can be classified into three different independent population groups, each containing a cluster of proliferating cells. Marker genes of group I, comprising *Wt1*-expressing EpiSC-1 and -7, have been reported in healthy adult mouse epicardium obtained by laser capture (*Bochmann et al., 2010*) and have been exclusively used in previous single-cell studies to identify epicardial cells (*Wagner et al., 2003*; *Bujak et al., 2009*; *Figure 1F*). Surprisingly, group I cells accounted for only 26% of the EpiSC fraction from the injured heart. Furthermore, the expression of several paracrine proangiogenic factors that were previously considered specifically derived from WT1⁺ epicardial cells (*Zhou et al., 2011*) was not limited to group I EpiSC, but was even higher in other EpiSC populations (*Figure 3E*). Thus, about two-third of all epicardial cells were *Wt1*-negative, but they are likely to be also involved in the secretion of paracrine factors. Furthermore, aCSC expressed numerous paracrine factors (*Figure 4H*). Therefore, secretion of paracrine factors appears to be a general feature of both epicardial and myocardial stromal cells, which all can contribute to cardioprotection. Group II, consisting of four cell populations, represented 40% of the EpiSC fraction and showed enriched expression of chemokines known to be involved in attraction of monocytes and neutrophils (*Figure 3E and F*). The expression of these chemokines was a general feature of EpiSC in comparison to aCSC (*Figure 4H*). This finding points to a role of epicardial cells in the modulation of the innate immune response post MI, as was already suggested with regard to adaptive immune regulation during the post-MI recovery phase (*Ramjee et al., 2017*).

Cells with the transcriptional profiles of group I and II populations were generally prevalent in EpiSC in comparison to aCSC (*Figure 4B*). Group I and II populations also showed the highest number of potential ligand–receptor interactions (*Figure 1I*). Interestingly, EpiSC-8, characterized by expression of *Wt1* and high transcript levels of HOX transcription factors, also highly expressed thymosin β4 (*Figure 3A and B*), which has tissue-regenerating properties (*Smart et al., 2011*). The pronounced expression of cardiogenic factors and contractile proteins in group I (*Figure 3A*) and the parallel expression of WT1 and thymosin β4 suggest that this cellular network has cardiogenic potential and was involved in the previously reported formation of cardiomyocytes from WT1⁺ cells after thymosin β4 stimulation (*Smart et al., 2011*; *Wang et al., 2021*). Interestingly, group I and II also

shared high expression of HIF-1-responsive glycolytic enzymes (*Figure 3C*), which again was a general feature of EpiSC compared to aCSC (*Figure 4G*). This is in support of the epicardium being a hypoxic niche, which was based on the immunostaining of dispersed HIF-1α-positive cells in the epicardium (*Kimura and Sadek, 2012*). Since WT1 expression is HIF-1-dependent (*Wagner et al., 2003*), this is consistent with the view that epicardial HIF-1 signaling is likely to be an important trigger in the ischemic heart to promote cardioprotection.

Group III comprised five cell populations accounting for 34% of all EpiSC. In contrast to group I, group III EpiSC were found localized throughout the activated epicardium (*Figure 2B*) and group III population identifiers also labeled stromal cells within the myocardium (*Figure 2B*). This finding is consistent with the transcriptional profile of group III EpiSC, which was quite similar to that of major aCSC and CSC populations (*Figure 4A and B*). This demonstrated that group III cells exhibit a fibroblast-like phenotype. In addition, group III EpiSC showed the lowest numbers of predicted cell–cell interactions (*Figure 1I*) and expressed a lower number of genes (*Figure 1H*), suggesting a less active cell state. Another remarkable feature of group III was the expression of a second set of cardiogenic factors (*Figure 3A*). However, these cardiogenic factors were prevalently expressed in aCSC (*Figure 4E*), which is consistent with the postulated cardiogenic potential of cardiac fibroblasts (*Furtado et al., 2014*).

That there are different and genuine cellular identities of EpiSC and aCSC is supported by CCA space alignment analysis and gene regulatory network analysis, which revealed distinct expression of epicardial vs mesenchymal/fibroblast genes (*Figure 4C*) and individual patterns in transcription factor activity (*Figure 4I and J*), respectively. That a direct comparison of cellular expression signatures between EpiSC and aCSC also revealed many transcriptional similarities was not surprising in view of their close developmental relationship (*Doppler et al., 2017*).

Histological observations by *Zhou et al., 2011* and *Quijada et al., 2020* found no evidence for migration of WT1[+] cells into the infarcted myocardium even after longer periods of time. That WT1[+] cells most likely do not migrate into the infarcted heart appears to be in contrast with a study using lentiviral labeling of epicardial cells after pericardial injection of virus expressing fluorescent proteins (*Gittenberger-de Groot et al., 2010*). Along the same line we have previously shown, that tracking of epicardial cells after labeling with fluorescently marked nanoparticles revealed migration into the injured heart (*Ding et al., 2016*). Since group II and III EpiSC constitute the majority of post-MI epicardial cells, it is well conceivable that populations of *Wt1*-negative cells were preferentially marked by the above-mentioned labeling techniques, and this may explain the reported migration into the injured myocardium. Our histological analysis of the epicardial post-MI multi-cell layer revealed that especially cells of group III populations are located in close proximity to the underlying myocardium (*Figure 2B*). In contrast, *Wt1*-expressing group I cells were located in the outmost part of the activated epicardium. Furthermore, our lineage tracing data suggest that until day 5 post MI, WT1[+] cells have not contributed to *Wt1*-negative group II and group III populations (*Figure 2E*). Therefore, group I cells appeared to be spatially restricted to the surface layers, while the major epicardial expansion seemed to be driven by *Wt1*-negative EpiSC of groups II and III with their respective proliferative cell (sub)populations. It is also conceivable that the fibroblast-like cells in group III represent myocardial stromal cells that have migrated into the epicardial multi-cell layer. The notion of a compartmentalization of the epicardial multi-cell layer is supported by the differential expression of hypoxia/HIF-1-responsive glycolytic genes among EpiSC populations, which is suggestive of an exposition to different local oxygen levels.

In summary, our study explored post-MI epicardial cell heterogeneity in the context of all cardiac stromal cells at an unprecedented cellular resolution. Important epicardial properties in post-MI wound healing/regeneration can now be attributed to specific cell populations. A deeper understanding of adult epicardial hierarchy may help decipher the signaling mechanisms by which individual epicardial cell populations interact to specifically stimulate cardiac repair processes originating in the epicardium.

## Materials and methods

**Key resources table**

*Continued*

| Reagent type (species) or resource | Designation | Source or reference | Identifiers | Additional information |
|---|---|---|---|---|
| Reagent type (species) or resource | Designation | Source or reference | Identifiers | Additional information |
| Strain, strain background (*Mus musculus*) | C57BL/6J | Janvier Labs | Strain name: C57BL/6JRj | |
| Strain, strain background (*M. musculus*) | STOCK *Wt1*<sup>tm2(cre/ERT2)Wtp</sup>/J | The Jackson Laboratory; PMID:18568026 | Stock no. 010912; RRID:IMSR_JAX:010912 | |
| Strain, strain background (*M. musculus*) | B6;129S6-*Gt(ROSA) 26Sor*<sup>tm14(CAG-tdTomato)Hze</sup>/J | The Jackson Laboratory | Stock no. 007908; RRID:IMSR_JAX:007908 | |
| Chemical compound, drug | Tamoxifen | Sigma-Aldrich | T5648 | |
| Chemical compound, drug | 7AAD | BioLegend | 420404 | (1:100) |
| Chemical compound, drug | Fixable Viability Dye eFluor 780 | eBioscience | 65–0865 | (1:100) |
| Other | Collagenase CLS II | Biochrom | C2-22 | Enzyme blend; now available from Sigma-Aldrich, C2-BIOC |
| Antibody | APC-conjugated rat monoclonal anti-CD31 | BioLegend | 102410, clone 390; RRID:AB_312905 | (1:100) |
| Antibody | PE-Cy7 rat monoclonal anti-CD45 | BD Biosciences | 552848, clone 30-F11; RRID:AB_394489 | (1:400) |
| Commercial assay or kit | RNAscope 2.5 HD Reagent Kit-RED | Advanced Cell Diagnostics | 322350 | |
| Sequence-based reagent | RNAscope Probe Mm-Msln | Advanced Cell Diagnostics | 443241 | |
| Sequence-based reagent | RNAscope Probe Mm-Wt1 | Advanced Cell Diagnostics | 432711 | |
| Sequence-based reagent | RNAscope Probe Mm-Cd44 | Advanced Cell Diagnostics | 479191 | |
| Sequence-based reagent | RNAscope Probe Mm-Ifit3 | Advanced Cell Diagnostics | 508251 | |
| Sequence-based reagent | RNAscope Probe Mm-Sfrp2 | Advanced Cell Diagnostics | 400381 | |
| Sequence-based reagent | RNAscope Probe Mm-Pcsk6 | Advanced Cell Diagnostics | 593361 | |
| Sequence-based reagent | RNAscope Probe Mm-Top2a | Advanced Cell Diagnostics | 491221 | |
| Sequence-based reagent | RNAscope Probe Mm-Dkk3 | Advanced Cell Diagnostics | 400931 | |
| Software, algorithm | CellRanger | 10x Genomics | | |
| Software, algorithm | Seurat | PMID:29608179 | RRID:SCR_007322 | |
| Software, algorithm | DoubletFinder | PMID:30954475 | RRID:SCR_018771 | |
| Software, algorithm | Scillus | https://github.com/xmc811/Scillus *Xu, 2021* | | |
| Software, algorithm | GOrilla | PMID:19192299 | RRID:SCR_006848 | |
| Software, algorithm | Velocyto.py | PMID:30089906 | RRID:SCR_018167 | |
| Software, algorithm | CellPhoneDB | PMID:32103204 | RRID:SCR_017054 | |
| Software, algorithm | SCENIC | PMID:28991892 | RRID:SCR_017247 | |

## Mice

All animal experiments were performed in accordance with the institutional guidelines on animal care and approved by the Animal Experimental Committee of the local government 'Landesamt für Natur, Umwelt und Verbraucherschutz Nordrhein-Westfalen' (reference number 81–02.04.2019. A181). The animal procedures conformed to the guidelines from Directive 2010/63/EU of the European Parliament on the protection of animals used for scientific purposes.

For this study, male C57BL/6J (Janvier Labs, Le Genest-Saint-Isle, France) and male tamoxifen-inducible *Wt1*-targeted (*Wt1$^{CreERT2}$Rosa$^{tdTomato}$*) mice were used. Inducible *Wt1$^{CreERT2}$Rosa$^{tdTomato}$* mice were generated by crossing the STOCK *Wt1$^{tm2(cre/ERT2)Wtp}$*/J strain (*Zhou et al., 2008*) (stock no. 010912; The Jackson Laboratory, Bar Harbor, USA) with the B6;129S6-*Gt(ROSA)26Sor$^{tm14(CAG-tdTomato)Hze}$*/J strain (stock no. 007908; The Jackson Laboratory, Bar Harbor, USA) followed by genotyping. Mice (body weight, 20–25 g; age, 8–12 weeks) used in this study were housed at the central animal facility of the Heinrich-Heine-Universität Düsseldorf (ZETT, Düsseldorf, Germany), fed with a standard chow diet, and provided tap water ad libitum.

## Animal procedures

MI followed by reperfusion was performed as previously described (*Bönner et al., 2012*). In brief, mice were anesthetized (isoflurane 1.5%) and the left anterior descending coronary artery (LAD) was ligated for 50 min followed by reperfusion. LAD occlusion was controlled by ST-segment elevation in electrocardiography recordings. Sham control animals underwent the surgical procedure without LAD ligation.

*Wt1$^{CreERT2}$*-mediated lineage tracing was performed as described previously (*Smart et al., 2011*). In brief, *Wt1$^{CreERT2}$Rosa$^{tdTomato}$* mice received tamoxifen injections (2 mg emulsified in sesame oil; i. p.) 5 and 3 days prior to MI to induce CreERT2 activity.

## Isolation of EpiSC, aCSC, and CSC

EpiSC and aCSC at day 5 after MI and control CSC from uninjured hearts at day 5 after sham surgery were isolated as previously described (*Owenier et al., 2020*). In brief, mice were sacrificed by cervical dislocation and hearts were excised for preparation of the aortic trunk in ice-cold phosphate-buffered saline (PBS). Isolated hearts were immediately cannulated and perfused with PBS (3 min), followed by perfusion with collagenase solution (8 min; 1200 U/ml collagenase CLS II (Biochrom, Berlin, Germany)) in PBS at 37°C.

EpiSC were simultaneously isolated from post-MI hearts by bathing the heart in its collagenase-containing coronary effluat while applying mild shear forces to the cardiac surface. The effluat was collected, centrifuged (300 g, 7 min), and cells were resuspended in PBS/2% fetal calf serum (FCS)/1 mM ethylenediaminetetraacetic acid (EDTA). Application of shear force and effluat collection were omitted for uninjured sham control hearts due to the absence of an activated, expanded epicardial layer. aCSC and CSC were isolated by mechanical dissociation of the digested myocardial tissue, followed by resuspension in Dulbecco's modified eagle medium (DMEM)/10% FCS. The cell suspensions were meshed through a 100 μm cell strainer and centrifuged at 55 ×g to separate cardiomyocytes from non-cardiomyocytes. The supernatants were again passed through a 40 μm cell strainer, centrifuged (7 min, 300 ×g), and cell pellets resuspended in PBS/2% FCS/1 mM EDTA. Cells were immediately stained for surface markers and applied to fluorescence-activated cell sorting.

## Fluorescence-activated cell sorting

Cell fractions (EpiSC, aCSC, CSC) were isolated as described above and stained at 4°C (15 min) with fluorochrome-conjugated antibodies against surface markers of endothelial cells (APC-conjugated rat monoclonal anti-CD31 [BioLegend, San Diego, USA]) and myeloid cells (PE-Cy7-conjugated rat monoclonal anti-CD45 [BD Biosciences, Franklin Lakes, USA]) in the presence of 7AAD (viability marker [BioLegend]). Sorting was performed with a MoFlo XDP flow cytometer (Beckman Coulter, Brea, USA), where dead cells (7AAD$^+$) and CD31$^+$/CD45$^+$ cells were excluded by gating on 7AAD$^-$/CD31$^-$/CD45$^-$ cells. Sorting of cells isolated from the *Wt1$^{CreERT2}$Rosa$^{tdTomato}$* mouse line followed the same gating strategy with one minor change. Fixable Viability Dye eFluor 780 (eBioscience, San Diego, USA) was used instead of 7AAD to avoid fluorescence spill-over.

## scRNAseq

The sorted single-cell suspensions were directly used for the scRNAseq experiments. scRNAseq analysis was performed by using the 10x Genomics Chromium System (10x Genomics, San Francisco, USA). Cell viability and cell number analysis were performed via trypan blue staining in a Neubauer counting chamber. A total of 2000–20,000 cells, depending on cell availability, were used as input for the single-cell droplet library generation on the 10x Chromium Controller system utilizing the Chromium Single Cell 3' Reagent Kit v2 according to manufacturer's instructions. tdTomato lineage tracing experiments were conducted utilizing the Chromium Single Cell 3' Reagent Kit v3 according to manufacturer's instructions. Sequencing was carried out on a HiSeq 3000 system (Illumina, San Diego, USA) according to manufacturer's instructions with a mean sequencing depth of ~90,000 reads/cell for EpiSC and ~70,000 reads/cell for aCSC. Differences in sequencing depth were necessary in order to achieve a similar sequencing saturation between all samples of ~70%.

## Processing of scRNAseq data

Raw sequencing data were processed using the 10x Genomics CellRanger software (v3.0.2) provided by 10x Genomics. Raw BCL-files were demultiplexed and processed to Fastq-files using the Cell-Ranger *mkfastq* pipeline. Alignment of reads to the mm10 genome and unique molecular identifier (UMI) counting were performed via the CellRanger *count* pipeline to generate a gene-barcode matrix.

The median of detected genes per cell was 3155 for EpiSC, 3265 for aCSC, and 2241 for CSC. The median of UMI counts per cell was 10,689 for EpiSC, 11,110 for aCSC, and 5879 for CSC. Mapping rates (reads mapped to the genome) were about 89% for EpiSC, 90.9% for aCSC, and 86.7% for CSC.

For tdTomato lineage tracing experiments, a custom reference, consisting of the mm10 genome and the full-length sequence of tdTomato, was generated via CellRanger *mkref*.

## Filtering and clustering of scRNAseq data

Further analyses were carried out with the Seurat v3.0 R package (*Butler et al., 2018*). Initial quality control consisted of removal of cells with fewer than 200 detected genes as well as removal of genes expressed in less than three cells. Furthermore, cells with a disproportionately high mapping rate to the mitochondrial genome (mitochondrial read percentages > 5.0 for EpiSC and aCSC, >7.5 for CSC) have been removed, as they represent dead or damaged cells. Normalization has been carried out utilizing SCTransform. Biological replicates have been integrated into one data set by identifying pairwise anchors between data sets and using the anchors to harmonize the data sets. Dimensional reduction of the data set was achieved by Principal Component Analysis (PCA) based on identified variable genes and subsequent UMAP embedding. The number of meaningful Principal Components (PCs) was selected by ranking them according to the percentage of variance explained by each PC, plotting them in an 'Elbow Plot' and manually determining the number of PCs that represent the majority of variance in the data set. Cells were clustered using the graph-based clustering approach implemented in Seurat v3.0. Doublet identification was achieved by using the tool DoubletFinder (v2.0.2) (*McGinnis et al., 2019*) by the generation of artificial doublets, using the PC distance to find each cell's proportion of artificial k nearest neighbors (pANN) and ranking them according to the expected number of doublets. Heat maps were generated using Morpheus (https://software.broad-institute.org/morpheus).

## Gene ontology enrichment analysis

Over-represented GO terms in differentially expressed genes between populations or among population groups were identified by using the Seurat wrapper Scillus (v0.5.0, https://github.com/xmc811/Scillus; *Xu, 2021*) and the gene ontology enrichment analysis and visualization (GOrilla) tool (*Eden et al., 2009*), respectively.

## RNA velocity analysis

The Python software velocyto.py (version 0.17) (*La Manno et al., 2018*) was run on the EpiSC count matrices and BAM files generated by CellRanger (see above) to predict and visualize future cell

states based on the ratio of unspliced and spliced mRNA read counts. For visualization, cell clusters, PCA, and UMAP data were imported by the Seurat analysis (see above).

### Cell cycle scoring

Scores for expression of cell cycle genes in EpiSC were assigned and classification of each cell in G1, G2/M, and S phase was predicted using the CellCycleScoring function implemented in Seurat.

### Cell–cell communication analysis

Cell–cell communication mediated by ligand–receptor complexes between EpiSC populations was analyzed using the tool CellPhoneDB v.2.0 (19) after mapping mouse genes to human orthologs.

### Canonical correlation analysis space alignment

A direct comparison of the EpiSC, aCSC, and CSC data sets was performed by Seurat's CCA alignment procedure (v2.3.4). Briefly, the top 600 variable genes were identified for each data set and subjected to a CCA. Herein, canonical correlation vectors were identified and aligned across data sets with dynamic time warping. After alignment, a single integrated clustering was performed, allowing for comparative analysis of cell populations across both cell fractions.

### Gene regulatory network analysis

Gene regulatory network reconstruction and cell-state identification in EpiSC, aCSC, and CSC data sets were performed using SCENIC (*Aibar et al., 2017*).

### RNA in situ hybridization

In situ detection of selected marker gene expression was performed by RNA in situ hybridization using the RNAscope 2.5 HD Reagent Kit-RED (Advanced Cell Diagnostics, Hayward, USA) (*Wang et al., 2012*) with RNAscope Probes targeting *Msln*, *Wt1*, *Cd44*, *Ifit3*, *Sfrp2*, *Pcsk6*, *Top2a*, and *Dkk3* mRNA sequences. Fresh frozen hearts (5 days after MI) were cut in 10 μm sections. Fixation and pretreatment of the cryosections were performed according to the manufacturer's instructions. The incubation time for the hybridization of RNAscope 2.5 AMP 5-RED was increased to 120 min to enhance the signal. For evaluation of the probe signal, the hybridized heart sections were examined under a standard bright-field microscope (BX61 Olympus, Hamburg, Germany) using a x20 objective. Images were processed for publication using ImageJ/Fiji (*Schindelin et al., 2012*).

### Statistics

Markers defining each cluster, as well as differential gene expression between different clusters, were calculated using a two-sided Wilcoxon Rank-Sum test that is implemented in Seurat.

## Acknowledgements

We thank K Raba (Institute of Transplantation Diagnostics and Cell Therapeutics, Heinrich-Heine-University Düsseldorf) for technical assistance with cell sorting by flow cytometry and J Schulze (Department for Genomics and Immunoregulation, Limes, Bonn) for helpful advice and discussions.

JS, JF, MG, and NG were supported by a grant of the German Research Council (DFG, SFB1116, project identifier: 236177352). CO was supported by the DFG-funded International Research Training Group 1902 (project identifier: 220652768), and this work was part of his PhD thesis. JH was supported by the Research Committee of the Medical Faculty of the Heinrich-Heine-University Düsseldorf (project identifier: 2018–12).

## Additional information

### Funding

| Funder | Grant reference number | Author |
| --- | --- | --- |
| Deutsche Forschungsgemeinschaft | 236177352 | Maria Grandoch<br>Norbert Gerdes<br>Jens W Fischer |

| | | Jürgen Schrader |
| Deutsche Forschungsge-meinschaft | 220652768 | Christoph Owenier |
| Medizinische Fakultät, Heinrich-Heine-Universität Düsseldorf | 2018-12 | Julia Hesse |

The funders had no role in study design, data collection and interpretation, or the decision to submit the work for publication.

### Author contributions
Julia Hesse, Conceptualization, Investigation, Visualization, Writing - original draft, Writing - review and editing; Christoph Owenier, Conceptualization, Investigation, Methodology, Writing - original draft; Tobias Lautwein, Data curation, Formal analysis, Investigation, Visualization, Writing - original draft; Ria Zalfen, Investigation, Visualization; Jonas F Weber, Formal analysis, Visualization; Zhaoping Ding, Christina Alter, Investigation; Alexander Lang, Conceptualization, Formal analysis, Writing - original draft; Maria Grandoch, Norbert Gerdes, Conceptualization, Resources, Writing - original draft; Jens W Fischer, Conceptualization, Resources, Funding acquisition; Gunnar W Klau, Resources, Formal analysis; Christoph Dieterich, Resources, Formal analysis, Visualization; Karl Köhrer, Resources, Data curation, Investigation, Writing - original draft; Jürgen Schrader, Conceptualization, Resources, Supervision, Funding acquisition, Writing - original draft, Writing - review and editing

### Author ORCIDs
Julia Hesse  https://orcid.org/0000-0001-8555-259X
Alexander Lang  https://orcid.org/0000-0001-6006-7109
Norbert Gerdes  https://orcid.org/0000-0002-4546-7208
Gunnar W Klau  http://orcid.org/0000-0002-6340-0090
Jürgen Schrader  https://orcid.org/0000-0002-7742-2768

### Ethics
Animal experimentation: All animal experiments were performed in accordance with the institutional guidelines on animal care and approved by the Animal Experimental Committee of the local government "Landesamt für Natur, Umwelt und Verbraucherschutz Nordrhein-Westfalen" (reference number 81-02.04.2019.A181). The animal procedures conformed to the guidelines from Directive 2010/63/EU of the European Parliament on the protection of animals used for scientific purposes.

### Decision letter and Author response
Decision letter https://doi.org/10.7554/eLife.65921.sa1
Author response https://doi.org/10.7554/eLife.65921.sa2

## Additional files
### Supplementary files
• Supplementary file 1. Average gene expression levels in EpiSC, aCSC, and CSC populations.

• Supplementary file 2. Genes with significantly enriched expression among EpiSC, aCSC, and CSC populations.

• Supplementary file 3. Selected ligand–receptor interactions between EpiSC populations as predicted by CellPhoneDB.

• Supplementary file 4. Average gene expression levels in subclusters of *Wt1*-expressing EpiSC populations.

• Supplementary file 5. Genes with significantly enriched expression among subclusters of *Wt1*-expressing EpiSC populations.

• Supplementary file 6. Genes with significantly enriched expression in group I and group II populations vs group III populations.

- Supplementary file 7. Average gene expression levels in clusters from CCA space alignment of EpiSC, aCSC, and CSC with separation in EpiSC, aCSC, and CSC, including p-values of differential expression.
- Supplementary file 8. Genes with significantly enriched expression among clusters from CCA space alignment of EpiSC, aCSC, and CSC.
- Transparent reporting form

### Data availability

ScRNAseq data have been deposited in the ArrayExpress database at EMBL-EBI (http://www.ebi.ac.uk/arrayexpress) under accession number E-MTAB-10035.

The following dataset was generated:

| Author(s) | Year | Dataset title | Dataset URL | Database and Identifier |
|---|---|---|---|---|
| Lautwein T, Schrader J | 2021 | Single-cell sequencing of epicardial cells and fibroblasts within the infarcted heart | https://www.ebi.ac.uk/arrayexpress/experiments/E-MTAB-10035 | ArrayExpress, E-MTAB-10035 |

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
