## [Decision Letter]

**Acceptance summary:**

The article by Hesse et al. defines heterogeneity of epicardial cells and fibroblasts in a murine model of cardiac injury to analyze the resulting populations through single cell RNA sequencing. Spatial confirmation of associated markers is performed using in-situ RNA hybridization and discussed in light of other investigations. There is extensive data that provides new insights into the heterogeneity of epicardial stromal and activated cardiac stromal cells post-injury.

**Decision letter after peer review:**

Thank you for submitting your article "Single-cell transcriptomics defines heterogeneity of epicardial cells and fibroblasts within the infarcted heart" for consideration by *eLife*. Your article has been reviewed by 3 peer reviewers, including Hina Chaudhry as the Reviewing Editor and Reviewer #1, and the evaluation has been overseen by a Senior Editor.

The reviewers have discussed their reviews with one another, this letter is to help you prepare a revised submission.

Essential revisions:

This manuscript is not suitable for publication in *eLife* in its current form. Essential revisions are required. All 3 reviewers have commented on the organization and flow of the manuscript which needs to be addressed.

In terms of title, the species needs to be identified according to the rules of *eLife*, thus the title should be changed to: Single-cell transcriptomics defines heterogeneity of epicardial cells and fibroblasts within the infarcted murine heart.

Other essential revisions include:

1) Please clarify your control and test conditions as Rev 2 requests. If the purity of the collected samples is described in a previous publication as noted by Rev 1, clarify that in the current manuscript.

2) Please address concerns regarding organization of the data and clarify cluster identities in Figure 1 as requested by Reviewer 3 and show heatmaps to show top 10 genes to see better segregation of genes across different clusters.

3) Please address the discrepancy brought up by Rev 3 between Figure 2 and 4: in Figure 2, the Hif1a/Hif1a-related and glycolysis related genes seem more highly expressed in 2 groups (WT1 and chemokine expressing cells). However, the feature plots and Dotplots supporting this need to be added to the figures. Show GO analysis with enrichment for Hif1a/glycolysis related genes in clusters vs. other clusters. It is shown that Hif1a related genes are expressed in clusters 1,7,8,4,2, and 11 but their lineage tracing analysis in Figure 4 suggests that Wt1+ cells (clusters 1,7) did not convert to any other cell type. Please address this discrepancy.

4) Please show experimental proof of dividing cells in Figure 2 with the use of (preferably) H3P to indicate mitosis or BrDU for DNA synthesis.

5) Please reorganize Results section: a lot of words are spent on cataloguing the genes identified in scRNA-seq instead of explaining their biological significance. Not until Figure 4 are the EpiSCs grouped into I, II, III based on their cell cycle active states where the whole manuscript begins to come into perspective. Then in Discussion, the authors continue to refer to groups I, II, III from Figures 1-3 where the 'grouping' concept was never introduced until Figure 4. Please implement the 'group' concept starting with Figure 1 so that it flows through the manuscript.

6) Provide further validations of genes defining cell clusters by in situ hybridization. It would be very interesting to understand more about where the different populations reside in the heart. Are they mixed? Are they at specific locations?

7) It would be helpful to see a comparison to EpiSc cells from the uninjured or sham control heart. If this is technically challenging, would it be feasible to pool 5-10 hearts to obtain adequate cell number?

8) It is surprising to see that TAM did not activate WT1+ cells at the chosen administration time window. Why was this time window chosen? Have the authors tested another time window as Day 5 is not really a 'regenerative' time window? If not, can you provide written justification? If 2/3 of all epicardial cells are WT1-negative, but are likely involved in paracrine signaling, is there another identifiable marker to label the WT1- cells?

*Reviewer #1 (Recommendations for the authors):*

In the manuscript "Single-cell transcriptomics defines heterogeneity of epicardial cells and fibroblasts within the infarcted heart", the authors isolated epicardial stromal cells (EpiSC) and cardiac interstitial/stromal cells (termed active CSCs) from the same I/R heart and identified transcriptionally distinct subpopulation of EpiSCs via 10x genomics technology. They also performed transcriptome profile comparison between EpiSCs and aCSCs. This manuscript shows rigorous scientific investigation. Their isolation protocol is supported by their previous publication. Method section documented in detail of step-wise QC process of bioinformatics analysis. In summary, the analysis identified 11 clusters of EpiSC, some of which overlap with the well-established epicardial marker WT1 with confirmed in situ anatomical localization. When compared to aCSC, the two groups showed clear different function/states as expected. In the lineage tracing model, RNA velocity predicts cell hierarchy, cell-cell communication between populations, as well as cell cycle activity. Overall this manuscript provides a ton of information that can be helpful to the field.

After I/R injury, the first 48-72hrs is the peak of inflammation followed by rapid cardiac function deterioration to one week after infarct. The chosen cell harvest time at 5 days post injury is in the middle of the deteriorating window, too early for any repair or regeneration to begin, especially without gene/cell intervention. My concern is that the EpiSC collected at this time is too early to be regenerative and therefore the manuscript missed peak of when epicardial cells are being most regenerative. This calls for a longitudinal comparison of EpiSCs post injury (eg. 5dpi, 10dpic, 30dpi).

In result section, authors spent a lot of words on cataloging the genes identified in scRNA-seq in stead of explaining the biological significance of them. It is until Figure 4 where the EpiSCs are grouped into I, II, III, based on their cell cycle active states where the whole manuscript began to come into perspective. Then in Discussion, the authors continued with group I, II, III terms talking about Figure 1-3 where the "grouping" concept was never introduced until Figure 4. It would be helpful if the "group" concept was implemented since Figure 1 and flows through the manuscript.

Cluster 3, being the 3rd largest EpiSC cluster, is shown to be "no GO enrichment found" in Figure 1E. Are these cells viable? please provide explanation of such observations.

As EpiSC can be a source of cell repair after injury, it'll be very interesting to see direct comparison of EpiSCs between MI and sham heart. RNA profiling of EpiSC in an injured heart may hold insights of their activity, migration, and paracrine signal during cardiac repair. Though the authors mentioned that isolating EpiSC from uninjured heart is technically challenging (very few cells), would it be feasible if pooling 5-10 hearts together to increase the cell number?

It is unexpected to see TAM did not activate WT1+ cells at the chosen administration time window. Why was such a plan chosen and have the authors tested other time window for TAM/harvest?

As mentioned in Discussion, about 2/3 of all epicardial cells were Wt1-negative but are likely to be involved in paracrine signaling. Is there an identifiable marker to label the 1/3 Wt1- cells?

*Reviewer #2 (Recommendations for the authors):*

1. It is important to explicitly define your control and test conditions. The schematic of Figure 1A needs to be explained in the text. What is the anticipated overlap in cell populations? What is the purity of isolation across these groups? This would need to be benchmarked. Does the isolation of aCSC have EpiSC contamination?

2. I would highly consider re-working the format of the manuscript. Namely, describing the changes across test conditions – which are unique/enriched to the injured heart, and then what changes are specifically enriched in EpiSC. For example, are EpiSC populations unique to the epidermal tissue but share transcriptomic similarity (i.e overlap of cells in Figure 3) or is there anticipated purity issues in isolation?

3. Please try to tease out differences between CSC and aCSC?

4. Walking a reader through the text using a cluster-based designation is difficult to follow because there is a need to constantly refer to a figure for orientation. I would suggest re-naming these with distinct population base names (e.g. Wt1-EpiSC7, Wt1-EpiSC1).

5. There is an attempt to perform EpiSC and aCSC comparison analysis in Figure 3, however without clarity on the anticipated purity, this is hard to interpret.

*Reviewer #3 (Recommendations for the authors):*

The following comments should allow the reader to better interpret or understand the significance of the work:

1) The current data is mostly based on scRNAseq analysis. There is confirmation by in situ hybridization analysis of only a few genes and information on the localization of cell populations within the heart superficially described and not being discussed.

2) In Figure 1, the authors name the clusters based on 2 genes and show Dotplots for 5 genes. It might help to check for more marker genes to confirm clusters identities, and show heatmaps (as supported by Seurat) to show top 10 genes and see better segregation of genes across different clusters.

3) The scRNAseq analysis broadly finds three groups of cells with 11 total clusters. It is possible that due to technical issues the resolution to cluster the cells may have been chosen such as to get 11 clusters. The three broad groups remain mostly conserved but 11 sub-groups might change if one changes minor parameters in the analysis.

4) In Figure 2, the Hif1a/Hif1a related and glycolysis related genes seem more expressed in 2 groups (WT1, and chemokine expressing cells). However, the feature plots and Dotplots supporting the same need to be added to the figures. It would be relevant to show GO analysis with enrichment for Hif1a/ glycolysis related genes in the clusters vs other clusters. The authors show that Hif1a related genes are expressed in clusters 1,7, 8, 4,2 and 11, but their lineage tracing analysis in Figure 4 suggests that Wt1+ cells (clusters 1,7) did not convert to any of the other cell type. It would be great to discuss this apparent discrepancy.

5) Groups defined as dividing cells were not corroborated experimentally by e.g. BrDu experiments (Figure 2).

6) In the cluster analysis for aCSC (Figure 3—figure supplement 1) similar to EpiSC, the cluster calling is based on only 2 genes. However, many of the clusters show overlapping gene expression patterns (cluster 4,5, clusters 7,10). The authors try to separate cluster 4, 5 based on differential expression patterns of Sca1, Smoc2, Sparcl1, but no-evidence for the same has been provided. Feature plots or Dot plots for such genes for Clusters 4,5 and 7,10 would improve cluster definition.

7) In Figure 3, authors club EpiSC and aCSC together, while in another analysis (Figure 3—figure supplement 5) they club EpiSC and CSC together. Such an approach would normalize the data for aCSC to EpiSC and CSC to EpiSC. However, this would not normalize the data for aCSC to CSC. The way the data is currently presented affects results interpretation. A better way to analyse this data would be to club all three datasets together and then perform the normalizations. If such an approach seems difficult it might be useful to perform clubbing of aCSC and CSC and analyse the levels of the genes in EpiSC in this condition.

8) The label transfer performed in Figure 4 (from analysis done in Figure 1) could also include the labels from the sub-clustering performed on Wt1+ EpiSC cells in Figure 2—figure supplement 1. This would help to understand better the RNA velocity data and help visualize and understand the lineage tracing to the more specific sub clusters- whether they originate from the more cell cycle associated what is the origin/fate of high protein expressing cells, etc.

9) The authors performed a ligand receptor analysis after the lineage tracing analysis, where they identify some pairs, which might be expressed in different groups. It would be useful to show at the analysis level the feature plot of the ligand receptors that indeed the expression levels of these pairs differs in the different groups. There is not confirmation by e.g. immunofluorescence that these proteins are expressed at different levels in different cells.

10) The manuscript seems to jump from EpiSC to comparisons of EpiSC with aCSC and again to analysis of EpiSC. The lineage tracing part of EpiSC of the manuscript could be clubbed after the clustering analysis of the EpiSCs. This would make all the EpiSC analysis stick together and the comparisons with aCSC at the end, making the flow of the story better.

---

## [Author Response]

Essential revisions:This manuscript is not suitable for publication in eLife in its current form. Essential revisions are required. All 3 reviewers have commented on the organization and flow of the manuscript which needs to be addressed.In terms of title, the species needs to be identified according to the rules of eLife, thus the title should be changed to: Single-cell transcriptomics defines heterogeneity of epicardial cells and fibroblasts within the infarcted murine heart.

The title has been changed accordingly.

[…] Reviewer #1 (Recommendations for the authors):In the manuscript "Single-cell transcriptomics defines heterogeneity of epicardial cells and fibroblasts within the infarcted heart", the authors isolated epicardial stromal cells (EpiSC) and cardiac interstitial/stromal cells (termed active CSCs) from the same I/R heart and identified transcriptionally distinct subpopulation of EpiSCs via 10x genomics technology. They also performed transcriptome profile comparison between EpiSCs and aCSCs. This manuscript shows rigorous scientific investigation. Their isolation protocol is supported by their previous publication. Method section documented in detail of step-wise QC process of bioinformatics analysis. In summary, the analysis identified 11 clusters of EpiSC, some of which overlap with the well-established epicardial marker WT1 with confirmed in situ anatomical localization. When compared to aCSC, the two groups showed clear different function/states as expected. In the lineage tracing model, RNA velocity predicts cell hierarchy, cell-cell communication between populations, as well as cell cycle activity. Overall this manuscript provides a ton of information that can be helpful to the field.After I/R injury, the first 48-72hrs is the peak of inflammation followed by rapid cardiac function deterioration to one week after infarct. The chosen cell harvest time at 5 days post injury is in the middle of the deteriorating window, too early for any repair or regeneration to begin, especially without gene/cell intervention. My concern is that the EpiSC collected at this time is too early to be regenerative and therefore the manuscript missed peak of when epicardial cells are being most regenerative. This calls for a longitudinal comparison of EpiSCs post injury (eg. 5dpi, 10dpic, 30dpi).

We agree that the post MI inflammation is a well-orchestrated process over time. As to the dynamics of the multi-cell layer of EpiSC we know that it is formed within the first several days after I/R and does not further increase in thickness after 4 days post MI in the murine heart (1). Furthermore, the re-expression of fetal epicardial genes as measure of epicardial activation peaks 1 and 5 days after MI and declines afterwards (2). Because of these reasons, we chose day 5 post I/R for the scRNAseq analysis of EpiSC. The rational for selecting day 5 post MI is now better explained in the revised manuscript (page 4 line 87-89). We certainly agree that it also would be interesting to analyze the composition of EpiSC populations at later time points in future studies.

In result section, authors spent a lot of words on cataloging the genes identified in scRNA-seq in stead of explaining the biological significance of them. It is until Figure 4 where the EpiSCs are grouped into I, II, III, based on their cell cycle active states where the whole manuscript began to come into perspective. Then in Discussion, the authors continued with group I, II, III terms talking about Figure 1-3 where the "grouping" concept was never introduced until Figure 4. It would be helpful if the "group" concept was implemented since Figure 1 and flows through the manuscript.

Thank you for this very helpful suggestion. We re-structured the manuscript, the grouping concept is now introduced in Figure 1.

Cluster 3, being the 3rd largest EpiSC cluster, is shown to be "no GO enrichment found" in Figure 1E. Are these cells viable? please provide explanation of such observations.

EpiSC-3 has a distinct gene expression pattern with multiple differentially expressed genes and cells are certainly viable. EpiSC-3 marker genes show enrichment of GO terms for Molecular Function and Cellular Component, however, no GO term for Biological Process was found (which was displayed in Figure 1E of the original manuscript). This finding seems to be consistent with the observation of a reduced number of expressed genes per cell in EpiSC-3 (Figure 1G), suggesting a rather quiescent cell state. Triggered by your question we have removed the presentation of manually selected Biological Process terms in original Figure 1E and replaced this panel by an unbiased visualization of enriched GO terms using the Seurat wrapper Scillus. This latter information is now in Figure 1—figure supplement 2 of the revised manuscript.

As EpiSC can be a source of cell repair after injury, it'll be very interesting to see direct comparison of EpiSCs between MI and sham heart. RNA profiling of EpiSC in an injured heart may hold insights of their activity, migration, and paracrine signal during cardiac repair. Though the authors mentioned that isolating EpiSC from uninjured heart is technically challenging (very few cells), would it be feasible if pooling 5-10 hearts together to increase the cell number?

Since the epicardium of healthy hearts is a monolayer, the isolation of these cells by enzymatic procedures is accompanied by a significant proportion of non-epicardial cells. We think that this contamination is the most critical issue rather than the low number of cells. Unfortunately, this problem cannot be overcome by pooling heart samples. However, in our CSC data set from uninjured hearts, a small yet defined population of epithelial/epicardial cells are visible as CSC-12, likely representing the epicardial monolayer. In CCA space alignment CSC-12 clustered together with cells of post-MI EpiSC-7 in CCA cluster A (Figure 4A and B). The direct comparison of CSC-12 and EpiSC-7 within CCA cluster A revealed differential expression of proliferation-associated and hypoxia-responsive genes pre and post MI and is now included in the manuscript (Figure 4—figure supplement 7D with source data).

It is unexpected to see TAM did not activate WT1+ cells at the chosen administration time window. Why was such a plan chosen and have the authors tested other time window for TAM/harvest?

Tamoxifen injection 5 and 3 days prior MI did induce expression of tdTomato in *Wt1*-expressing cells as expected. The question which we intended to answer was, whether *Wt1*-expressing cells may contribute to the *Wt1*-negative populations of the thickened epicardium that is formed during the acute injury response. We have better clarified this in the text (Results section “Spatial mRNA expression of EpiSC population identifiers”).

As mentioned in Discussion, about 2/3 of all epicardial cells were Wt1-negative but are likely to be involved in paracrine signaling. Is there an identifiable marker to label the 1/3 Wt1- cells?

The 2/3 of *Wt1*-negative EpiSC display variable degrees of similarity to myocardial aCSC. While we observed distinct epicardial/myocardial signatures in relative gene expression (Figure 4 C-H), we did not identify any marker that clearly distinguishes Wt1-negative EpiSC and aCSC. Since the spatial distribution of *Wt1*-negative EpiSC in the epicardial post-MI multi-cell layer was verified by in situ hybridization, this most likely indicates that they are derived from cells already present in the epicardial monolayer of the uninjured heart, or might have migrated from the myocardium into the epicardial layer after injury.

Reviewer #2 (Recommendations for the authors):1. It is important to explicitly define your control and test conditions. The schematic of Figure 1A needs to be explained in the text. What is the anticipated overlap in cell populations? What is the purity of isolation across these groups? This would need to be benchmarked. Does the isolation of aCSC have EpiSC contamination?

A detailed protocol and comprehensive validation data of our cell isolation procedure for EpiSC, aCSC and CSC have been previously published (4), including analyses of cell yield, viability, and purity. We have clarified this in the text and extended the explanation of the schematic in Figure 1A (Results section “ScRNAseq of post-MI stromal cells”).

The aCSC fraction contains a small fraction of EpiSC, as was observed by positive immunostaining of WT1 protein in a small quantity of short-term cultured aCSC before (4). In the present study, aCSC‑6 was identified as residual EpiSC according to high gene expression of WT1+ together with other epicardial/epithelial genes (Figure 4—figure supplement 1D). As is now better explained in the Results section “Post-MI aCSC in comparison to EpiSC”, the EpiSC layer is likely not completely removed from the myocardium during the cell isolation procedure. However, changing the protocol to extended digestion times would amplify the probability of non-epicardial cells in the EpiSC fraction.

In the revised manuscript we have rearranged data presentation – as requested by the other reviewers – and have carefully revised the text for clarity to avoid misunderstandings.

2. I would highly consider re-working the format of the manuscript. Namely, describing the changes across test conditions – which are unique/enriched to the injured heart, and then what changes are specifically enriched in EpiSC. For example, are EpiSC populations unique to the epidermal tissue but share transcriptomic similarity (i.e overlap of cells in Figure 3) or is there anticipated purity issues in isolation?

The epicardium of adult healthy hearts is a monolayer of epithelial cells. Myocardial ischemia is a prerequisite for generation of stromal cells in the epicardium, forming the EpiSC multilayer. As mentioned above in the response to reviewer 1, we identified epithelial cells of the epicardial monolayer in CSC-12 of our CSC data set from the uninjured heart. There was no other CSC population with expression of epicardial/epithelial markers (Figure 4—figure supplement 4D and E). In the CCA space alignment of cell fraction data sets, CSC-12 overlapped with the epithelial cells of the post-MI heart (EpiSC-7) in CCA cluster A. We have included a comparative analysis of CSC-12 and EpiSC-7 revealing the MI-induced transcriptional changes in these epithelial cells (Figure 4—figure supplement 7D with source data).

3. Please try to tease out differences between CSC and aCSC?

Since there are already elaborate scRNAseq studies specifically addressing the activation of CSC after MI published (5,6). We therefore refrained from comparing CSC and aCSC again and instead focused on the novel scRNAseq data of the post-MI epicardium.

4. Walking a reader through the text using a cluster-based designation is difficult to follow because there is a need to constantly refer to a figure for orientation. I would suggest re-naming these with distinct population base names (e.g. Wt1-EpiSC7, Wt1-EpiSC1).

Thank you for this very helpful comment. We have re-structured and revised the manuscript and now use the classification of EpiSC populations in group I, II, and III throughout the Results and Discussion sections as was also suggested by reviewer 1 (2.). We are confident that this has significantly improved the general readability of our text.

5. There is an attempt to perform EpiSC and aCSC comparison analysis in Figure 3, however without clarity on the anticipated purity, this is hard to interpret.

We have made an attempt to clarify the text in the Result sections “ScRNAseq of post-MI stromal cells” and “Post-MI aCSC in comparison to EpiSC” regarding the issue of purity (see also answer to 1.).

Reviewer #3 (Recommendations for the authors):The following comments should allow the reader to better interpret or understand the significance of the work:1) The current data is mostly based on scRNAseq analysis. There is confirmation by in situ hybridization analysis of only a few genes and information on the localization of cell populations within the heart superficially described and not being discussed.

We have now included a text passage specifically addressing the location of EpiSC populations in the post-MI epicardium in the Discussion section.

2) In Figure 1, the authors name the clusters based on 2 genes and show Dotplots for 5 genes. It might help to check for more marker genes to confirm clusters identities, and show heatmaps (as supported by Seurat) to show top 10 genes and see better segregation of genes across different clusters.

Heat maps visualizing the expression of top 10 marker genes are now included for EpiSC (Figure 1—figure supplement 3), aCSC (Figure 4—figure supplement 3), and CSC (Figure 1—figure supplement 6).

3) The scRNAseq analysis broadly finds three groups of cells with 11 total clusters. It is possible that due to technical issues the resolution to cluster the cells may have been chosen such as to get 11 clusters. The three broad groups remain mostly conserved but 11 sub-groups might change if one changes minor parameters in the analysis.

You are absolutely correct that changing the resolution parameter during cluster identification will lead to a different number of detected clusters. We have discussed this issue and finally decided for the chosen resolution because this resulted in the most biological meaningful results. We have tested different resolution parameters during data analysis, but this either leads to over-clustering, resulting in multiple biological non-meaningful clusters, or to under-clustering, leading to significant loss of cellular heterogeneity.

4) In Figure 2, the Hif1a/Hif1a related and glycolysis related genes seem more expressed in 2 groups (WT1, and chemokine expressing cells). However, the feature plots and Dotplots supporting the same need to be added to the figures. It would be relevant to show GO analysis with enrichment for Hif1a/ glycolysis related genes in the clusters vs other clusters. The authors show that Hif1a related genes are expressed in clusters 1,7, 8, 4,2 and 11, but their lineage tracing analysis in Figure 4 suggests that Wt1+ cells (clusters 1,7) did not convert to any of the other cell type. It would be great to discuss this apparent discrepancy.

To resolve this issue, we have added supporting violin plots to the Figure (now Figure 3). GO enrichment analysis of the genes with significantly higher expression in group I and II populations *vs*. group III populations (Supplementary file 6) indeed yielded terms associated with the hypoxia/HIF-mediated metabolic switch towards glycolysis, such as Glycolytic Process (FDR q-value: 3.05E-02), Glucose Metabolic Process (FDR q-value: 3.22E-02), and Pyruvate Biosynthetic Process (FDR q-value: 2.66E-02). This information is now included in the Results (section “Expression of cardiomyogenesis-associated genes, HIF-1-responsive genes, and paracrine factors”).

There is no discrepancy at all in the observations that these populations share enriched expression of HIF-1-responsive genes and that the WT1- among them do not derive from the Wt1+ ones. The highly conserved HIF-1 response is due to the tight regulation of HIF-1α protein stability only transient and strongly depends on environmental conditions (acute hypoxia). Therefore, if cells are located in a microenvironment with low oxygen levels, HIF-1 will be induced and its target genes will be expressed, but the cells don’t need to be of the same lineage. We have clarified this in the Results (section “Expression of cardiomyogenesis-associated genes, HIF-1-responsive genes, and paracrine factors”) and the Discussion.

5) Groups defined as dividing cells were not corroborated experimentally by e.g. BrDu experiments (Figure 2).

Van Wijk et al. have already nicely shown the presence of proliferating cells in the epicardial post-MI multi-cell layer by BrdU incorporation (7). We have now added this information to the text (Results section “Spatial mRNA expression of EpiSC population identifiers”).

6) In the cluster analysis for aCSC (Figure 3—figure supplement 1) similar to EpiSC, the cluster calling is based on only 2 genes. However, many of the clusters show overlapping gene expression patterns (cluster 4,5, clusters 7,10). The authors try to separate cluster 4, 5 based on differential expression patterns of Sca1, Smoc2, Sparcl1, but no-evidence for the same has been provided. Feature plots or Dot plots for such genes for Clusters 4,5 and 7,10 would improve cluster definition.

As already mentioned above (2.), we have now included a heat map of aCSC top 10 marker genes (Figure 4—figure supplement 3) that confirms cluster identities. Additionally, we included feature plots of selected marker genes visualizing the segregation of aCSC-4, 5 and aCSC-7,10, respectively (Figure 4—figure supplement 1E).

7) In Figure 3, authors club EpiSC and aCSC together, while in another analysis (Figure 3—figure supplement 5) they club EpiSC and CSC together. Such an approach would normalize the data for aCSC to EpiSC and CSC to EpiSC. However, this would not normalize the data for aCSC to CSC. The way the data is currently presented affects results interpretation. A better way to analyse this data would be to club all three datasets together and then perform the normalizations. If such an approach seems difficult it might be useful to perform clubbing of aCSC and CSC and analyse the levels of the genes in EpiSC in this condition.

Thank you for making us aware of this issue. To avoid potential misinterpretation due to separate normalization, we replaced the individual CAA space alignments (EpiSC and aCSC or EpiSC and CSC) by a combined analysis of all three fractions. This information is now displayed in Figure 4.

8) The label transfer performed in Figure 4 (from analysis done in Figure 1) could also include the labels from the sub-clustering performed on Wt1+ EpiSC cells in Figure 2—figure supplement 1. This would help to understand better the RNA velocity data and help visualize and understand the lineage tracing to the more specific sub clusters- whether they originate from the more cell cycle associated what is the origin/fate of high protein expressing cells, etc.

Thank you for this helpful suggestion. We have added the label transfer of the *Wt1*-expressing subclusters (Figure 2—figure supplement 3).

9) The authors performed a ligand receptor analysis after the lineage tracing analysis, where they identify some pairs, which might be expressed in different groups. It would be useful to show at the analysis level the feature plot of the ligand receptors that indeed the expression levels of these pairs differs in the different groups. There is not confirmation by e.g. immunofluorescence that these proteins are expressed at different levels in different cells.

Feature plots visualizing the expression levels of selected receptor-ligand pairs across different EpiSC populations are now included as Figure 1—figure supplement 4.

10) The manuscript seems to jump from EpiSC to comparisons of EpiSC with aCSC and again to analysis of EpiSC. The lineage tracing part of EpiSC of the manuscript could be clubbed after the clustering analysis of the EpiSCs. This would make all the EpiSC analysis stick together and the comparisons with aCSC at the end, making the flow of the story better.

Very good point, thank you! We have re-structured the manuscript (as was also suggested by reviewer 1), so that now the comparison of EpiSC, aCSC, and CSC is at the end in Figure 4 and the lineage tracing results are incorporated in Figure 2.

References:

1. Ding Z, Temme S, Quast C, Friebe D, Jacoby C, Zanger K, et al. Epicardium-Derived Cells Formed After Myocardial Injury Display Phagocytic Activity Permitting in vivo Labeling and Tracking. Stem Cells Transl Med. 2016 May;5(5):639–50.

2. Zhou B, Honor LB, He H, Ma Q, Oh J-H, Butterfield C, et al. Adult mouse epicardium modulates myocardial injury by secreting paracrine factors. J Clin Invest. 2011 May 2;121(5):1894–904.

3. Kocabas F, Mahmoud AI, Sosic D, Porrello ER, Chen R, Garcia JA, et al. The Hypoxic Epicardial and Subepicardial Microenvironment. J of Cardiovasc Trans Res. 2012 Oct 1;5(5):654–65.

4. Owenier C, Hesse J, Alter C, Ding Z, Marzoq A, Petzsch P, et al. Novel technique for the simultaneous isolation of cardiac fibroblasts and epicardial stromal cells from the infarcted murine heart. Cardiovasc Res. 2020 Apr 1;116(5):1047–58.

5. Farbehi N, Patrick R, Dorison A, Xaymardan M, Janbandhu V, Wystub-Lis K, et al. Single-cell expression profiling reveals dynamic flux of cardiac stromal, vascular and immune cells in health and injury. Morrisey E, Dietz HC, editors. *eLife*. 2019 Mar 26;8:e43882.

6. Forte E, Skelly DA, Chen M, Daigle S, Morelli KA, Hon O, et al. Dynamic Interstitial Cell Response during Myocardial Infarction Predicts Resilience to Rupture in Genetically Diverse Mice. Cell Reports. 2020 Mar 3;30(9):3149-3163.e6.

7. Wijk B van, Gunst QD, Moorman AFM, Hoff MJB van den. Cardiac Regeneration from Activated Epicardium. PLOS ONE. 2012 Sep 20;7(9):e44692.